# Assessment of Maturity of Plum Samples Using Fourier Transform Near-Infrared Technique Combined with Chemometric Methods

**DOI:** 10.3390/foods12163059

**Published:** 2023-08-15

**Authors:** Marietta Fodor, Zsuzsa Jókai, Anna Matkovits, Eszter Benes

**Affiliations:** Department of Food and Analytical Chemistry, Institute of Food Science, Hungarian University of Agriculture and Life Sciences, Villányi út 29–43, 1118 Budapest, Hungary; jokaine.szatura.zsuzsanna@uni-mate.hu (Z.J.); matkovits.anna@phd.uni-mate.hu (A.M.); benes.eszter.luca@uni-mate.hu (E.B.)

**Keywords:** plum, FT-NIR, PLSR, classification methods, maturity

## Abstract

The FT-NIR technique was used for rapid and non-destructive determination of plum ripeness. The dry matter (DM), titratable acidity (TA), total soluble solids (TSS) and calculated maturity index (MI: TSS/TA) were used as reference values. The PLS correlations were validated via five-fold cross-validation (RMSECV for different parameters: DM: 0.66%, *w*/*w*; TA = 0.07%, *w*/*w*; TSS = 0.72%, *w*/*w*; MI = 1.39) and test set validation (RMSEP for different parameters: DM: 0.65%, *w*/*w* TA = 0.07%, *w*/*w*; TSS = 0.61%, *w*/*w*; MI = 1.50). Different classification algorithms were performed for TA, TSS and MI. Linear, quadratic and Mahalanobis discriminant analysis (LDA, QDA, MDA) were found to be the best sample detection methods. The accuracy of the classification methods was 100% for all investigated parameters and cultivars.

## 1. Introduction

Plum has been cultivated for about 5000 years, as a stone fruit with a wide range of uses and high nutritional value. Several plum species exist in different areas of the planet, such as American (*Prunus nigra*, *Prunus americana*), Chinese (*Prunus simonii*) and Japanese (*Prunus salicina*) species [1]. The climatic conditions in Hungary are favorable for the cultivation of the most common varieties of European plum (*Prunus domestica* L.), which has high ecological tolerance. Succulent, fleshy and juicy plums are low in calories and saturated fats, but rich in vitamin C, K and A, minerals (magnesium, potassium, calcium, etc.) and other various biogenic components, such as anthocyanins and polyphenol-type compounds that contribute to preservation of human health (Table 1) [2,3,4].

It is well known that nutritional properties of fruit (dry matter content, sugar content, acidity, pH, polyphenol profile, antioxidant capacity, etc.) change during the ripening process, so monitoring these components can be used to estimate their ripeness.

The changes in the concentrations of typical phenolic components (epicatechin, catechin, B2, neochlorogenic acid and chlorogenic acid and other typical phenolic acids, such as gallic acid and caffeic acid) depending on the harvest time of *Prunus salicina* fruits were investigated by Cabrera-Bañegil et al. [5] based on the excitation–emission spectra of the samples. Parallel factor analysis (PARAFAC) and unfolded partial least-squares (U-PLS) data processing were applied to quantify the polyphenol compounds. It was concluded that the fluorescence spectra do not allow separate determination of epicatechin and catechin, nor neochlorogenic acid and chlorogenic acid. The prediction models were suitable for quantification of the components and the results obtained were in agreement with the values determined via high-performance liquid chromatography with fluorescence detection (HPLC-FLD). However, it was observed that the total amount of catechin and epicatechin and chlorogenic and neochlorogenic acids decreased with maturation. Using the same measurement and data processing technique, Monago-Maraña et al. [6] found that the chlorophyll content of plums can be a good indicator of the ripening process. Based on their results, fluorescence fingerprint combined with second-order calibrations is suitable for monitoring chlorophyll content in plum fruit. Fluorescence spectroscopy combined with chemometric analysis was used by Monago-Maraña et al. [7] to distinguish Japanese ‘Angeleno’ plum cultivars according to harvest date. Based on the polyphenol content, the classification models predicted the stage of maturity with acceptable accuracy. In addition, the calibration models obtained with partial least-square (PLS) regression also gave good results for the individual quantification of polyphenols like neochlorogenic acid and epicatechin.

In another study on this topic [8], changes in total soluble solids content (TSS) and firmness during the storage of plum samples were investigated, and classification for varietal identification was performed with a hand-held near-infrared (NIR) instrument using partial least-squares discriminant analysis (PLS-DA) evaluation. Two different measurement techniques were used by the authors, a handheld micro-electro-mechanical system spectrophotometer—MEMS—and a diode-array Vis-NIR spectrophotometer. Since the measurement range of the two instruments is different (1600–2400 nm for the former and 515–1650 nm for the latter) and relatively narrow, the obtained prediction models were not very good. The root mean square error of cross-validation (RMSECV) for TSS determination was 1.11% in the case of Vis-NIR and 1.39% in the case of MEMS (the measurement range was 8.30–19.60%). At the same time, firmness (1.77) and RMSECV values (2.76) (the measuring range was 1.93–16.12) were found using these two measurement techniques, respectively [8,9].

Louw and co-workers [10] used Fourier transform near-infrared (FT-NIR) reflectance spectroscopy to develop a multivariate prediction model for the TSS, total titratable acidity (TA), sugar-to-acid ratio (TSS/TA), hardness and weight of three South African plum cultivars. The measurements were carried out for two years during 7 weeks of the ripening period with mixed validation results. It was found that among the parameters studied, the TSS models had the best statistical characteristics. However, it was observed that the statistical performance of the models also varied depending on the variety, with better predictive models obtained for the varieties ‘Pioneer’ and ‘Laetitia’ than for ‘Angeleno’. The best model was found to predict TSS (for the varieties ‘Pioneer’ and ‘Laetitia’) (square of coefficient of determination for the cross-validation or test set validation: Q^2^ = 0.817–0.959; RMSEP = 0.453–0.610%, Brix). The reliability of the results is supported by the very large number of samples (*n* > 1000) and the fact that the sample was taken over two years. The FT-NIR technique was applied by Costa et al. [11] to study *Prunus salicina*, L. and *Prunus domestica* samples (48 samples) for TSS determination (5–15%) and pH (2.72–3.84) using different variable selection procedures (interval partial least-squares regression—iPLS, genetic algorithm—GA, successive projection algorithm—SPA and ordered predictor selection—OPS). Spectra were recorded from five different points of the samples using a diffuse reflectance measurement setup. For TSS, PLS regression without variable selection has the most favorable statistical properties, with a root mean square error of prediction (RMSEP) of 0.45%. For pH, PLS-GA has the most favorable RMSEP (0.07).

According to another publication, the diffuse reflectance NIR method was used for the non-destructive examination of the browning of fruit flesh during storage. During the qualitative tests, the Mahalanobis distances discriminate analysis (DA) and the backpropagation–artificial neural networks (BP-ANN) were used to detect brown and non-brown flesh. Using the BP-ANN method, 100% accuracy was achieved for brown and non-brown samples [12].

The changes in TSS, TA, pH, firmness, TSS/TA and flesh color (L*, a*, b*) parameters of ‘Friar’ plums (*Prunus domestica*, Friar) during cold storage were investigated by Li et al. [13]. Measurements were performed for 28 days in the wavelength range 638–986 nm using the Vis/NIR technique. Among the investigated parameters, a highly favorable statistical correlation was obtained only for TSS (Q^2^ = 0.9456, RMSEP = 0.456).

The sugar profile of the juice of *Prunus domestica* plum varieties ‘Vânăt de Italia’, ‘Stanley’ and ‘Tuleu Gras’ was investigated by Vlaic et al. [14] during the ripening process. Samples of different maturity stages were measured using the Fourier transform mid-infrared (FT-MIR) technique. It was found that during the ripening process, the fructose level of the samples varied between 0.26 and 3.73%, the glucose level between 1.43 and 1.10% and the sucrose content between 0.01 and 10.19%. The best estimation result, as expected given the concentrations, was obtained for sucrose (Q^2^ = 0.97; RMSEP = 0.57).

Point spectroscopy offers an interesting way to monitor the ripening process of fruits and vegetables. It provides the sum signals of attenuation, i.e., absorption and scattering. The scattering properties of a tissue influence the detected signal. However, in the case of European plums, the scattering rate varies during the fruit’s growth. Consequently, the apparent absorption changes, which upsets the relationship between apparent absorption and the attributes under investigation, so this method is not recommended for plum samples [15].

The NIR technique does not only allow the estimation of well-known quality parameters. In combination with appropriate chemometric techniques, it can also be used to detect *Monilinia fructigena* infection. Using independent linear discriminant analysis (LDA) prediction, plum samples not yet showing signs of *M. fructigena* infection can be clearly identified based on their spectral characteristics [16].

For fruit quality control, shape, size, skin color and general appearance are the basic external quality parameters while TSS, TA, TSS/TA, pH, starch and sugar content, carotenoids, sugar, ascorbic acid, total flavonoids, total phenolic, antioxidant activity and flesh firmness are indicators of internal quality properties [17].

The aim of our research was to develop a fast and non-destructive method for determining the maturity state of plum samples based on the most obvious quality parameters (TSS, TA, TSS/TA and pH).

We further aimed to develop classification models by evaluating reference data and varieties using different chemometric techniques. These models can be used for direct monitoring of fruit ripeness to quickly select the right quality and variety at the processing site.

## 2. Materials and Methods

### 2.1. Materials

Investigations were carried out on the two most typical varieties of the Szabolcs-Szatmár-Bereg region of Hungary, namely, *Prunus domestica* cv. ‘Elena’ (38 samples, marked E) and *Prunus domestica* cv. ‘Stanley’ (30 samples, marked S) (Figure 1) [18,19]. The ‘Stanley’ cultivar ripens at the end of August with a dark blue and very ash skin. They are colored early, so the color is not an objective parameter for maturity. It is suitable for consumption or processing only when fully ripened. The ‘Elena’ cultivar ripens at the end of September. Its color and bloom are similar to those of the ‘Stanley’ cultivar; it is a very sweet and aromatic fruit. It has a higher average sugar content than the ‘Stanley’ cultivar. The ripening stages of these two varieties were studied over two years (2021–2022). In 2021, the samples (both varieties) were harvested in mid-August and early September, and in 2022 in late August and mid-September. Immature and mature samples of both varieties were tested.

### 2.2. Methods

The dry matter content (DM), titratable acidity (expressed as malic acid) (TA), total soluble solids content (Brix°) (TSS), pH and the maturity index calculated from the sugar/acid ratio (MI = TSS/TA) were determined to determine the ripening stage. Three parallel measurements were made for each parameter.

#### 2.2.1. Reference Methods

The reference methods have already been described in detail in our previous work [20].

#### 2.2.2. FT-NIR Measurements

The FT-NIR spectra were recorded, and the data were processed using a Bruker MPA FT-NIR instrument (BRUKER, Ettlingen, Germany). The diffuse reflectance spectra were recorded with a resolution of 16 cm^−1^, and the final spectral image was obtained by averaging 32 sub-spectra.

A rotating quartz sample holder (Ø 85 mm) was used to provide the largest possible surface area. Spectra were taken from the original fruit sample; no sample preparation was applied. Five spectra were recorded for each sample. The evaluation was performed using the average of the parallel spectra [20].

Spectral data were evaluated using OPUS 7.2 (Bruker, Ettlingen, Germany). Chemometric evaluation (quantitative model building, classification models) was performed using the Classification Learner application of Matlab 2022b (MathWorks, Natick, MA, USA) and Unscrambler 10.4 (CAMO, Oslo, Norway) software.

#### 2.2.3. Chemometric Methods

##### Principal Component Analysis—PCA

PCA is an unsupervised pattern recognition technique. Several algorithms, including Singular Value Decomposition (SVD) and NIPALS, can be used to find the principal components. The major practical difference between the two methods is that, unlike for NIPALS, in SVD, the scores are scaled so that the sum of squares of the scores of each component is equal [21].

##### Partial Least-Squares Regression—PLSR

The most important metrics of the PLSR model are the coefficient of determination (R^2^ for calibration, Q^2^ for validation), the root mean square error (RMSECV for cross-validation and RMSEP for test validation), the number of the latent variables (PLS factors), the value of the residual prediction deviation (RPD) and the bias. The calculation of these parameters is based on the following mathematical relationships:(1)$$RMSECV;RMSEP=\sqrt{\frac{1}{N}\sum_{i=1}^{N}{\left(y_{i}^{m}-y_{i}^{p}\right)}^{2}}$$
(2)$$bias=\frac{1}{N}\sum_{i=1}^{N}\left(y_{i}^{m}-y_{i}^{p}\right)$$
(3)$$RPD={\left(1-Q^{2}\right)}^{-0.5}$$
where

RMSECV or RMSEP: root mean square error of cross-validation or test validation (the unit of measurement is the same as that of the estimated parameter);

$y_{i}^{m}$: measured (reference) value of the ith component;

$y_{i}^{p}$: estimated or predicted value of the ith component;

N = number of samples;

Q^2^ = squared coefficient of determination for validation.

During model building, the main objective is to ensure that the characteristic parameters take the optimal value. This means that R^2^ and Q^2^ should be as close to 1 as possible, and that RMSECV and RMSEP and the associated bias (which should be no more than one-tenth of the mean squared error) should be as low as possible. Residual prediction deviation (RPD) is a model-specific quality parameter calculated from R^2^ and Q^2^, but as it is not an independent datum, not everyone uses it. Nevertheless, experience shows that this parameter helps to qualify the model. If the RPD > 3 (this means Q^2^ > 0.89), the model is considered excellent for quantitative evaluation [22,23].

The maximum number of PLS factors was set to ten depending on the number of samples tested to avoid the under- or overfitting.

Data preprocessing was performed in order to reduce variations in the spectral data (e.g., variations due to sample thickness and light scattering) using the following methods: straight-line subtraction (SLS) standard normal variate (SNV), multiplicative scatter correction (MSC), derivatives (first and second derivative, FD and SD) or a combination of SLS, SNV and MSC with FD or SD algorithms [24,25,26].

The PLSR models were validated through random three-fold cross-validation and test set validation. For the latter, the dataset was split in a 70:30 ratio (48 samples for the training set, 20 samples for the test set). The training and test samples were randomly selected. However, the allocation of the samples also took into account that both datasets should cover the full range of measurement parameters.

##### Classification Models

Different supervised learning algorithms were performed (discriminant analysis; decision trees; nearest-neighbor method; multilayer perceptron neural network, naïve Bayes; partial least-squares discriminant analysis; random forest; support vector machine) using spectral data with different classification criteria based on maturity of the sample. All models were validated using a random five-fold cross-validation procedure [27].

The model performance was evaluated based on commonly used classification metrics, such as sensitivity, specificity, precision and accuracy. These were calculated based on the number of true-positive (T_P_), true-negative (T_N_), false-positive (F_P_) and false-negative (F_N_) values, correctly classified results N_CORR_ and total results N_TOT_ using the following equations [28,29]:(4)$$Sensitivity=\frac{T_{P}}{T_{p}+F_{N}}$$
(5)$$Specificity=\frac{T_{N}}{T_{N}+F_{P}}$$
(6)$$Precision=\frac{T_{P}}{T_{p}+F_{P}}$$
(7)$$Accuracy=\frac{N_{CORR}}{N_{TOT}}$$

## 3. Results

### 3.1. Reference Data

The measurement results provided by the reference methods and the reference data available in the literature are summarized in Table 2. The measurement results refer to fresh fruit in all cases.

### 3.2. NIR Spectra Analysis

In comparing the spectra of the two cultivars, characteristic differences can be seen in the raw spectra (Figure 2) in the 5000–3800 cm^−1^ wavenumber range. A vibrational transition of the bonds and functional groups of sugars and organic acids can be detected in this range.

The signal from surface scattering (wax layer) and surface inhomogeneity can be eliminated via spectra derivation. Therefore, a qualitative comparison of the spectra on the first or second derivatives is appropriate. The difference in the 12,500–11,000 cm^−1^ range is related to the color of the samples and therefore should be ignored. Differences between the first-derivative spectra of cultivars can be observed in several wavenumber ranges (Figure 2). The difference observed in the range of 5000–3800 cm^−1^ of the raw spectra is even more evident in the first-derivative spectra. Furthermore, a peak shift in the 6500–5500 cm^−1^ region can be observed, which is related to the fiber content and refers to the different quality of the fiber [35,36].

The vibrational regions of the important parameters for the mature state—DM, TA and TSS—are marked in different colors in Figure 2 [37].

### 3.3. Chemometric Assessment

Chemometric assessment—including PCA, PLS and classification methods—were performed using Unscrambler 10.4 and the Classification Learner application of Matlab software.

#### 3.3.1. Principal Component Analysis—PCA

##### Spectral Data

PCA was performed using the SVD algorithm with 10 principal components to cover the variability of the samples.

A five-fold random validation was used as a control.

The following criteria were set in the PCA algorithm: the ratio of calibrated to validated residual variances should be 0.5, the ratio of validated to calibrated residual variance should be 0.75, and the residual variance increase limit should be 6%.

It can be concluded that based on the residual variance (Figure 3), the first three principal components explain the majority of the variance of the traits (98%).

In examining the three principal components, it can be seen there are some samples that fall outside the 95% confidence interval (ellipse) (Figure 4a,b).

To determine whether these samples are indeed spectral outliers, the F residual and Hotelling’s T^2^ relationship need to be examined (Figure 5). Hotelling’s T^2^ statistic describes the distance from the model center to the principal components.

It was observed that samples with high F residuals but low Hotelling’s T^2^, i.e., samples lying in region B of the plot, are poorly described by the model (samples E36, E38 and S21). Since high residual variance is associated with less important spectral regions, these samples need not be excluded from the model.

Samples with high Hotelling’s T^2^ but low F residuals, i.e., samples lying in region C of the plot, are well described by the model (sample S16). However, such samples may be influential to the model.

The classic outlier samples, which are located in region D, have high F residuals and high Hotelling’s T^2^ and influence the model. In our case, there were no such samples.

##### Reference Data

Score plot analysis was used to examine the impact of the reference data. Correlation loadings were calculated for each variable for the displayed principal components, as shown in Figure 6. The plot contains two ellipses indicating the magnitude of the variance being considered. The outer ellipse is the unit circle and indicates the explained variance of 100%. The inner ellipse indicates 50% of explained variance. The effect of the maturity index was not examined, as it is a calculated value. Based on the correlation loadings, it is possible to determine which parameters determine each principal component.

The TSS value is located at the 100% ellipse. The DM is close to the 100% ellipse and is on the positive side, as is the TSS value. In contrast, the TA is in the negative range as expected, as the titratable acidity and the total soluble solids content of the sample move in the opposite direction during ripening.

The pH point is in the positive range between the 50 and 100% ellipse. This is interesting because pH and TA change in opposite directions due to the nature of pH. However, it should be noted that pH is not a robust property and therefore not the best indicator of the state of maturity. The first principal component is clearly determined by the TSS value (PC-1 = 95%), while PC-2 is determined by DM (PC-2 = 4%). Titratable acidity has a small effect (PC-3 = 1%) (Figure 6b).

#### 3.3.2. PLSR Results

The PLSR models with the best statistical parameters are summarized in Table 3. The maximum Mahalanobis limit was 0.5.

Prediction models for Q^2^ > 0.900 and RPD > 3 were obtained for both cross-validation and test set validation for all parameters tested except for dry matter content.

A good model has also been established for the statistical properties of pH, but this relationship is only apparently correct. The pH values had a very narrow measurement range (2.95–3.99) during the tests, often only varying within one hundredth of a percent. It was concluded that this property is not sensitive enough to determine maturity; therefore, this model is not included in Table 3.

#### 3.3.3. Classification Methods

Classification models were tested according to maturity (mature/immature) and variety (Stanley/Elena). Taking into account the literature data and our measured results (Table 2), the TA and TSS ranges for the mature and immature stages were established as shown in Table 4. The classification of mature/immature samples was checked using ANOVA, Tukey’s post hoc test and Duncan’s new multiple range test for DM, TA, TSS and MI = TSS/TA (S1).

As a first step of the spectrum-based classification, ten principal component PCA data reductions were performed. Classification models were then established for the mature/immature samples based on each of the investigated properties (Table 5).

The best-performing relationships were identified via different types of discriminant analysis (linear, quadratic, Mahalanobis).

Taking into account the reference data and the spectral data, it can be concluded that the same samples were classified as mature based on the parameters TA, TSS and TSS/TA by the two datasets, except for one sample. This one sample was S21, whose TSS fell within the range established for mature samples but whose TA was 1.16% by weight, i.e., outside the range of ≤1.00% established by us. Therefore, this sample was proved to be immature based on TA.

Unfortunately, as described earlier, DM and pH values are not considered good indicators of maturity status. The relationship between DM values and maturity is not fully clear.

The same problem arises for pH values for the PLSR model. Therefore, no classification was performed for either DM or pH.

Using the classification models applied to the spectra, mature/immature samples could be classified with 100% accuracy.

For the different varieties, the classification of Stanley and Elena samples based on spectra analysis alone was also perfect, with 100% accuracy.

Ripeness is generally characterized by values of TA, TSS and MI. Measuring the hardness of the flesh of the fruit could be added in order to refine the evaluation of ripeness (Appendix A).

An extension of the sampling period is also recommended. A longer sampling period would result in establishing categories of immature, medium mature, mature and over mature. The meaning of ‘ripe’, which is different in the sense of commerce, storage and canning, could also be specified.

## 4. Conclusions

A non-destructive FT-NIR technique was successfully developed for the determination of the dry matter (interval: 16.72–24.28%, *w*/*w*, RMSECV: 0.66%, *w*/*w*, RMSEP: 0.65%, *w*/*w*) titratable total acidity (interval 0.50–1.70%, *w*/*w*; RMSECV = 0.07%, *w*/*w*, RMSEP = 0.0.07%, *w*/*w*), total soluble solids (interval 7.90–19.4%, *w*/*w*; RMSECV = 0.72%, *w*/*w*, RMSEP = 0.61%, *w*/*w*) and maturity index (interval 5.20–37.60; RMSECV = 1.39, RMSEP = 1.50) of plum samples.

The methods were validated with five-fold random cross-validation and test set validation. In the case of test set validation, the number of samples in the calibration and validation datasets was 48:20.

Based on our reference values, the typical concentration intervals for the mature and immature samples were determined for each parameter tested.

Various methods of pattern recognition and discriminant analysis, decision trees, the nearest-neighbor method, multilayer perceptron neural networks, naïve Bayes, partial least-squares discriminant analysis, random forests and support vector machines were investigated. Among the investigated methods, LDA, quadratic discriminant analysis (QDA) and Mahalanobis discriminant analysis (MDA) produced the most favorable results.

The classification of mature and immature samples was performed with 100% accuracy for the TA, TSS and MI = TSS/TA parameters.

Consequently, it can be stated that the determination of the mature state can be made based on TA, TSS and TSS/TA values.

The relationship between the studied parameters and plum varieties was also investigated. It was found that the models developed were able to distinguish between the Elena and Stanley varieties with 100% accuracy. This is very important as the varieties have more or less different characteristics, so their uses may also differ. In summary, prediction models provide rapid quality control, even on the production line. Classification models based on spectral databases allow for rapid, even automatic, identification of varieties and maturity status.

## Figures and Tables

**Figure 1 foods-12-03059-f001:**
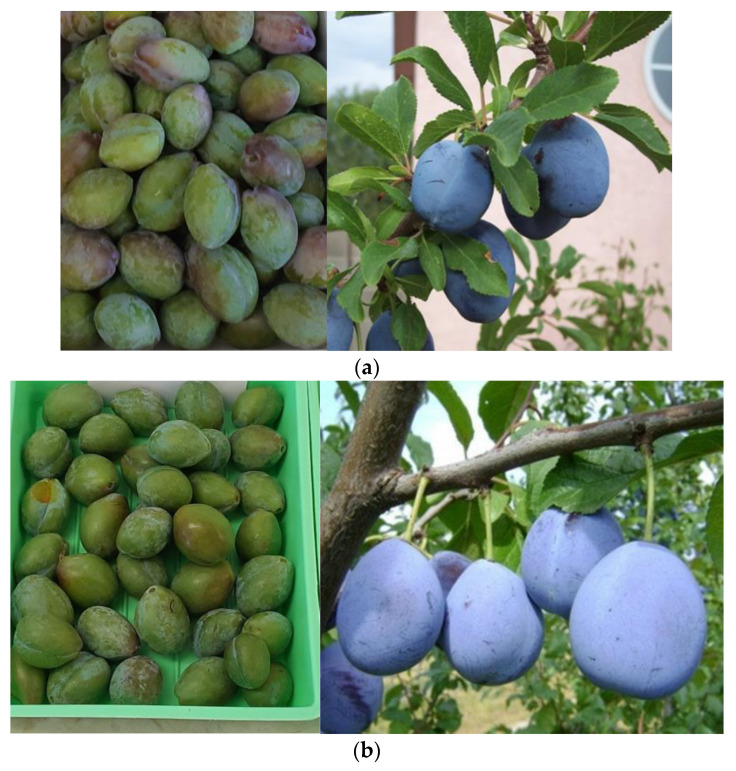
The investigated plum varieties at different stages of maturity ((**a**); immature and mature *Prunus domestica* cv. Stanley; (**b**) immature and mature *Prunus domestica* cv. Elena).

**Figure 2 foods-12-03059-f002:**
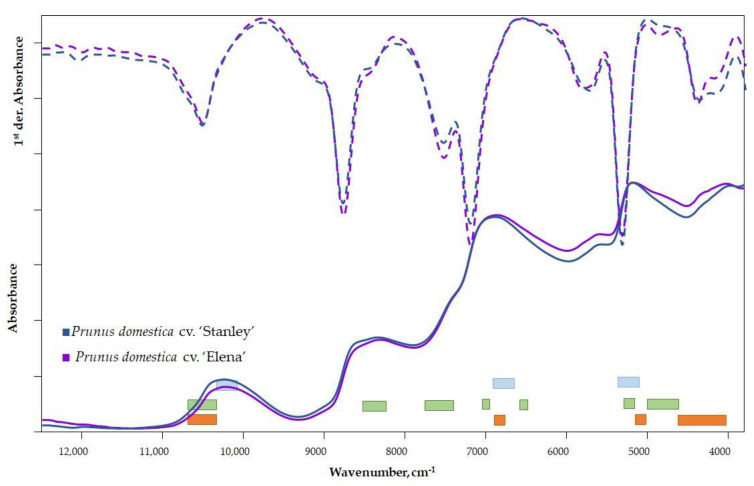
Raw and first-derivative spectra of plum varieties (■ TSS, ■ TA, ■ water).

**Figure 3 foods-12-03059-f003:**
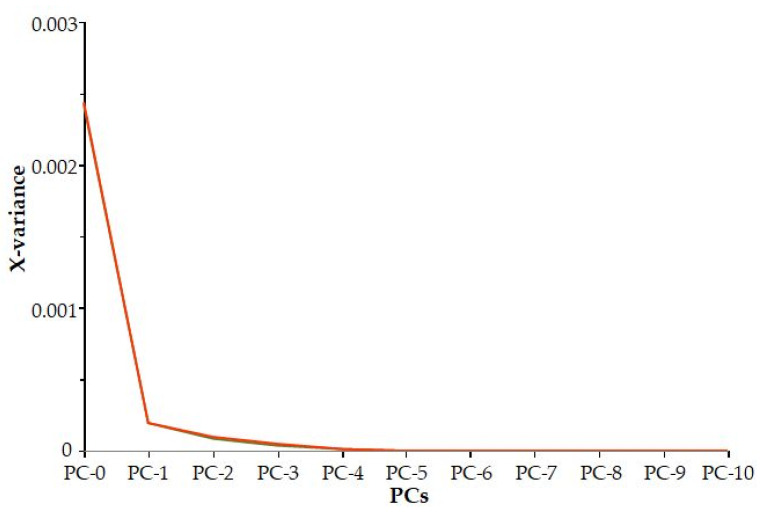
Residual variance.

**Figure 4 foods-12-03059-f004:**
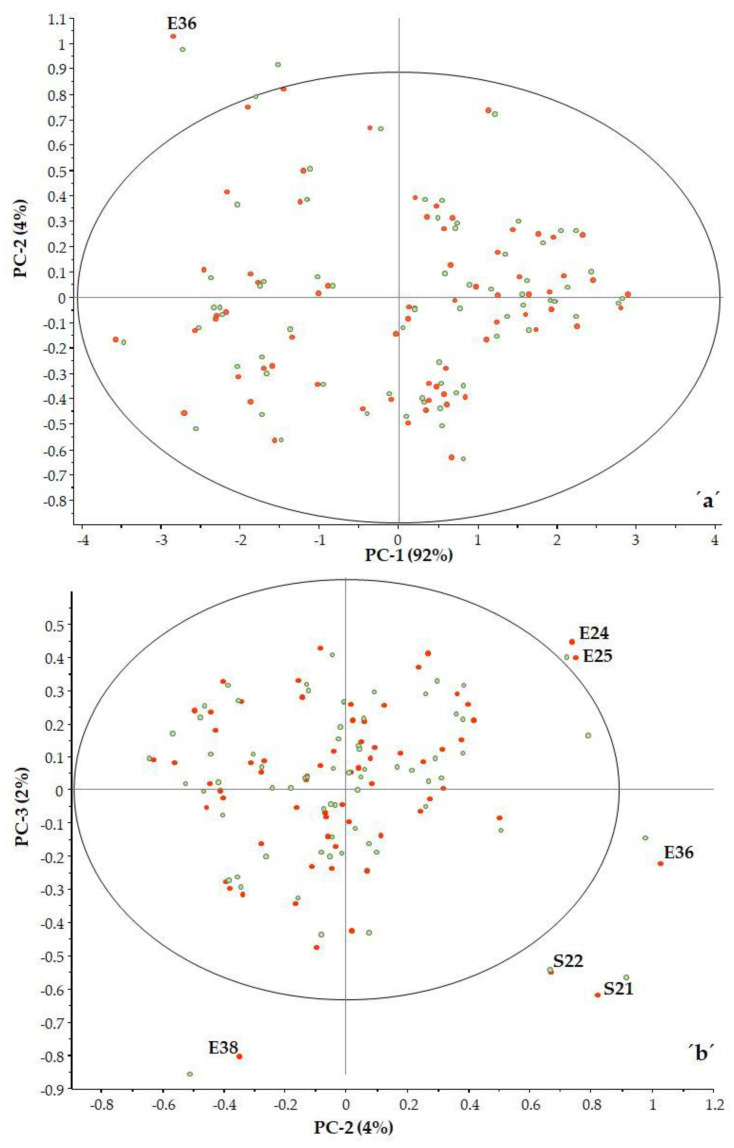
PCA analysis PC2 vs. PC1 (**a**) and PC3 vs. PC2 (**b**) (■ calibration, ■ validation).

**Figure 5 foods-12-03059-f005:**
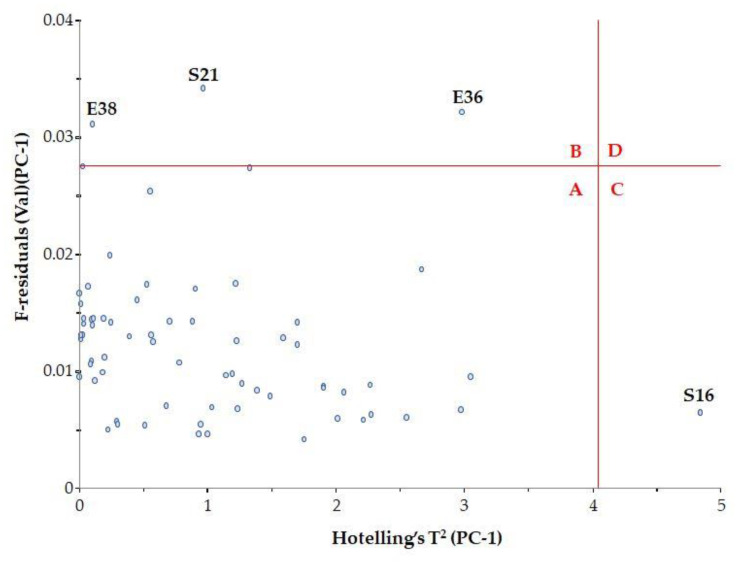
Influence plot with Hotelling’s T^2^.

**Figure 6 foods-12-03059-f006:**
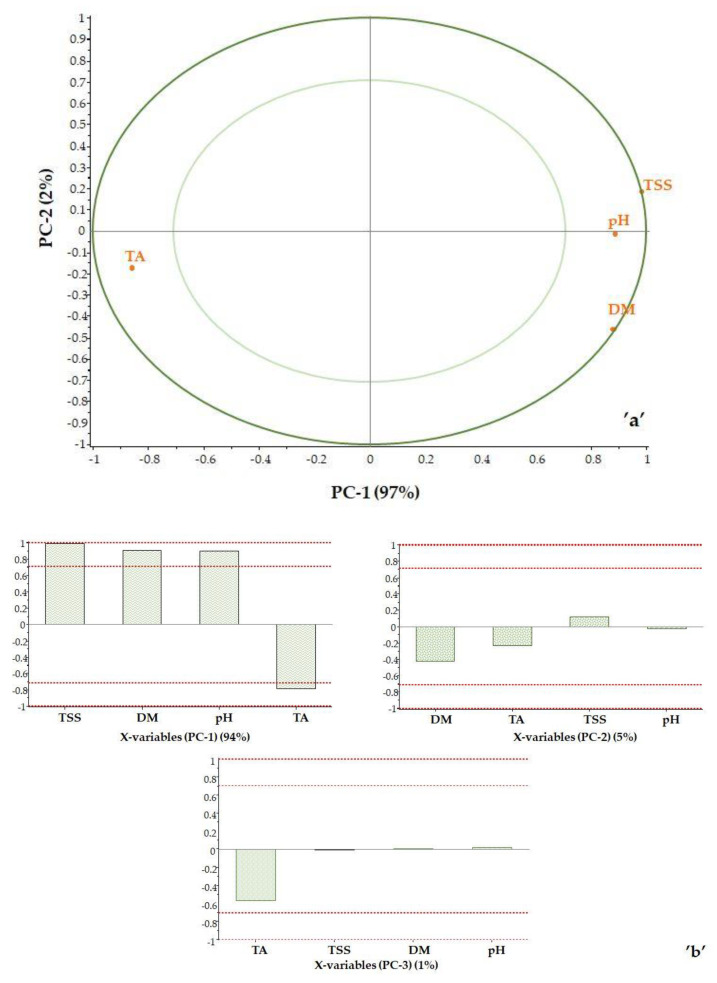
Correlation loadings: PC2 vs. PC1 (**a**) and the effect of properties on the principal components (**b**).

**Table 1 foods-12-03059-t001:** Energy and macronutrient content of fresh plums (per 100 g sample) [3,4].

Components	Amount, Unit	Components	Amount, Unit
Water *	84 g	Energy **	38 kcal/159 kJ
Ash *	0.37 g	Protein *	0.7 g
Carbohydrate **	9.6 g	Total lipid (fat) *	0 g
Fiber, total dietary *	2.1	Starch *	0 g
Total sugar *	9.92 g	Vitamin content *	
Sucrose *	1.57 g	Thiamin	0.028 mg
Glucose *	5.07 g	Riboflavin	0.026 mg
Fructose *	3.07 g	Niacin	0.417 mg
Galactose *	0.14 g	Pantothenic acid	0.135 mg
Mineral content *		Vitamin C	9.5 mg
Ca	6 mg	Vitamin B6	0.029 mg
Fe	0.17 mg	Vitamin B12	0
Mg	7 mg	Vitamin A	17 μg/345 IU
P	16 mg	Vitamin E (α-tocopherol)	0.26 mg
K	157 mg	Vitamin K (phylloquinone)	
Zn	0.1 mg		6.4 μg
Cu	0.057 mg		
Mn	0.052 mg		
β-Carotene *	190 μg	Lutein + zeaxanthin *	73 μg
SFA **	17 mg	MUFA **	134 mg
16:0	14 mg	16:1	2 mg
18:0	3 mg	18:1	132 mg
PUFA 18:2 **	44 mg		

* Measured; ** calculated; SFA—total saturated fatty acids; MUFA—total monounsaturated fatty acids; PUFA—total polyunsaturated fatty acids.

**Table 2 foods-12-03059-t002:** Results of the reference measurements and reference data available in the literature.

Parameters	Measured Concentration Range	Reference Data for Ripe Fruit	Reference
DM; % *w*/*w*	16.32–28.61	13.47–16.49	[30,31,32]
TA; % *w*/*w*	0.50–1.70	0.91–0.98	
TSS; % *w*/*w*	7.90–19.40	9.77–16.38	
TSS/TA	5.20–38.80	13.22	
pH	2.95–3.99	2.8–3.4	[33,34]

**Table 3 foods-12-03059-t003:** The best PLSR models for cross-validation and test set validation.

Parameter	Sample Number	Calibration		PC	Cross-Validation				Data Preprocessing
		R^2^	RMSEE		Q^2^_cv_	RMSECV	RPD_CV_	Bias	
DM	68	0.925	0.54	9	0.865	0.66	2.73	−0.0119	SNV + FD
TA		0.976	0.05	8	0.950	0.07	4.48	0.0023	
TSS		0.986	0.42	10	0.951	0.72	4.51	0.0223	
MI = TSS/TA		0.987	0.79	9	0.955	1.39	4.70	0.0031	
**Parameter**	**Sample * Number**	**Calibration**		**PC**	**Test Validation**				**Data Preprocessing**
		**R^2^**	**RMSEE**		**Q^2^_p_**	**RMSEP**	**RPD_P_**	**Bias**	
DM	48/20	0.946	0.50	9	0.882	0.65	2.92	−0.0413	SNV + FD
TA		0.977	0.06	8	0.949	0.07	4.43	−0.0042	
TSS		0.991	0.35	10	0.965	0.61	5.33	−0.0105	
MI = TSS/TA		0.976	1.12	9	0.951	1.50	4.70	−0.086	

* Calibration/test sample.

**Table 4 foods-12-03059-t004:** Classification data of mature and immature plums.

Parameter	Mature	Immature
TA; %, *w*/*w*	≤1.00	>1.00
TSS; %, *w*/*w*	≥10.0	<10.0
TSS/TA	≥10.0	<10.0

**Table 5 foods-12-03059-t005:** Parameters of classification models based on spectral data.

	**Mature**	**Immature**	**Accuracy**	**Precision**		**Sensitivity**		**Specificity**	**Classification** **Method**
			**%**						
**Titratable acidity (TA) %, *w*/*w***									
Mature	47	0	100.0	100		100		100	MDA QDA
Immature	0	21		100		100		100	
**Total Soluble Solid Content (TSS) %, *w*/*w***									
Mature	48	0	100	100		100		100	LDA, MDA; QDA
Immature	0	20		100		100		100	
**Maturity index = TSS/TA**									
Mature	48	0	100	100	0	100	0	100	LDA, MDA; QLDA
Immature	0	20		100	0	100	0	100	
**Cultivars**									
	**Elena**	**Stanley**	**Accuracy**	**Precision**		**Sensitivity**		**Specificity**	**Classification** **method**
Elena	38	0	97.06	100		100		100	LDA, MDA; QDA
Stanley	0	30		100		100		100	

LDA—linear discriminant analysis; MDA—Mahalanobis discriminant analysis; QDA—quadratic discriminant analysis.

## Data Availability

The data presented in this study are available on request from the authors.

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
