# Peer review of "Assessment of Maturity of Plum Samples Using Fourier Transform Near-Infrared Technique Combined with Chemometric Methods"

_foods, 2023, doi:10.3390/foods12163059_

Round 1

Reviewer 1 Report

It is suggested that add the descriptions of the PCA part in the abstract correspond to the keywords.

Although the author introduces a lot of work in the introduction, but the lack of logic and there are too many repetitive descriptions, suggestions are streamlined. When quoting the results of others, there are more abbreviations and full names missing or repeated. and there is a lack of elaboration on the differences between author 's work and previous work.

Line 56: The abbreviation 'PLS' should be placed before 'regression'.

Line 63 - Line 65: Sentences lack proper punctuation, and the first occurrence of an abbreviation in line 65 needs to be given the full name.

Line 77: The full name of TSS has already appeared in Line 63 and does not need to be brought up again. The ' FT-NIR ' of 87 lines and 76 lines are the same problem.

Line 85: What is the meaning of Q2, please give a description for the first occurrence.

Line 109: The full name of FT-MIR should be given. As well as the LDA in Line 124.

Line 261: What is the basis for the number of PLS factors selected as 10. Please give an explanation.

Line 269 - Line 271: What method was used to divide the dataset.

Line 353: Principal Components doesn 't need capital letters.

Line 442 - Line 443: There is no need to give the full name here again.

Minor editing of English language required

Author Response

Assessment of maturity of plum samples using FT-NIR technique combined with chemometric methods

by

Marietta Fodor, Zsuzsa Jókai, Anna Matkovits, Eszter Benes

Response to Reviewer 1

Thank you for your time to review and thank you for your comment

It is suggested that add the descriptions of the PCA part in the abstract correspond to the keywords.

PCA has been removed from the keywords

Although the author introduces a lot of work in the introduction, but the lack of logic and there are too many repetitive descriptions, suggestions are streamlined. When quoting the results of others, there are more abbreviations and full names missing or repeated. and there is a lack of elaboration on the differences between author 's work and previous work.

Thanks for the comment. The Introduction section has been revised a bit; the repetitions (Table 2) have been deleted.

Line 56: The abbreviation 'PLS' should be placed before 'regression'.

Thank you, it was corrected

Line 63 - Line 65: Sentences lack proper punctuation, and the first occurrence of an abbreviation in line 65 needs to be given the full name.

Thank you, it was corrected

Line 77: The full name of TSS has already appeared in Line 63 and does not need to be brought up again. The ' FT-NIR ' of 87 lines and 76 lines are the same problem.

Thanks, it was corrected, but FT-NIR appears first in the text on line 76, that's why it's written out

Line 85: What is the meaning of Q2, please give a description for the first occurrence.

Square of the coefficient of determination for the cross- or test set validation

Line 109: The full name of FT-MIR should be given. As well as the LDA in Line 124.

Thank you, it was corrected

Line 261: What is the basis for the number of PLS factors selected as 10. Please give an explanation.

The number of PLS factors is strongly influenced by the number of samples tested. Experience from our work so far shows that ten is a value at which overfitting can be avoided. However, it should also be noted that there is a risk of underfitting if too few factors are used.

Line 269 - Line 271: What method was used to divide the dataset.

The training and test samples were randomly selected. However, the allocation of the samples also took into account that both data sets should cover the full range of measurement parameters

Line 353: Principal Components doesn 't need capital letters.

Thank you, it was corrected

Line 442 - Line 443: There is no need to give the full name here again.

Thank you, it was corrected

Yours sincerely,

Marietta Fodor

Reviewer 2 Report

I liked your research work, I also appreciated the statistical analysis you have used. However, from the title I was expecting to find more practical indication on how to determine the harvest time using FT-Nir technique.

In your introduction are listed several "basic" study reporting nutritional properties of fruit (polyphenol, -catechin, epicatechin, procyanidin etc) and their antioxidant capacity. The bibliography is accurate and reported the results obtained by several researchers. Some of them also consider Flesh firmness, that is a practical parameters used. Flesh firmness is also considered an important parameter for fruit ripening assessment and several Authors you cited consider it.  It is a parameter difficult to be determined by NIR technique but is one used in the practice in most of the fruits.

In your study only TA and TSS are considered. 

I ask to the Authors about Fruit Flesh firmness since , although important, is difficult to be assessed with precision with non-destructive methods.

If eventually they have data it will appropriate to report them

Author Response

Assessment of maturity of plum samples using FT-NIR technique combined with chemometric methods

by

Marietta Fodor, Zsuzsa Jókai, Anna Matkovits, Eszter Benes

Response to Reviewer 2

Thank you for your time to review and thank you for your comment.

I liked your research work, I also appreciated the statistical analysis you have used. However, from the title I was expecting to find more practical indication on how to determine the harvest time using FT-Nir technique.

In your introduction are listed several "basic" study reporting nutritional properties of fruit (polyphenol, -catechin, epicatechin, procyanidin etc) and their antioxidant capacity. The bibliography is accurate and reported the results obtained by several researchers. Some of them also consider Flesh firmness, that is a practical parameters used. Flesh firmness is also considered an important parameter for fruit ripening assessment and several Authors you cited consider it.  It is a parameter difficult to be determined by NIR technique but is one used in the practice in most of the fruits.

In your study only TA and TSS are considered.

Thank you for your time to review and thank you for your comment on the fruit flesh test.

We did not test the flesh of the fruit because the primary objective was to develop a method that could also monitor the ripening process.

We only distinguished between two categories: ripe and unripe. Within this, we could refine what the canning industry would like to use the fruit for. The concept of ripeness is different if, for example, it is used to make drinking juice, and different if, for example, it is processed into preserves or jam.

Our work could be the basis for further categories, but this will of course require more samples and more measurements to be added to the current database

 Yours sincerely,

Marietta Fodor

Reviewer 3 Report

This manuscript tried to construct a model using the FT-NIR technique to assess the maturity of plum samples. The following suggestions could improve this manuscript

1.    Figure 1 Please provide a photo of the two varieties of plum samples at the two different ripening stages.

2.    Please use the model constructed by the two years of data (2021-2022) in this manuscript to check whether it could be successfully applied in this year's plum maturity detection.

Author Response

Assessment of maturity of plum samples using FT-NIR technique combined with chemometric methods

by

Marietta Fodor, Zsuzsa Jókai, Anna Matkovits, Eszter Benes

Response to Reviewer 3

Thank you for your time to review and thank you for your comment.

This manuscript tried to construct a model using the FT-NIR technique to assess the maturity of plum samples. The following suggestions could improve this manuscript

  1. Figure 1 Please provide a photo of the two varieties of plum samples at the two different ripening stages.

Thank you for the suggestion. Please find attached photos of the immature and mature stage. The manuscript has been supplemented with a photograph of the unripe fruit

                         Stanley                                                       Elena

  1. Please use the model constructed by the two years of data (2021-2022) in this manuscript to check whether it could be successfully applied in this year's plum maturity detection.

Unfortunately, we cannot yet apply the estimation models to this year's plum crop because the plums are later in ripening. The first ripe fruits can be harvested in late August, early/mid-September (depending on the variety).

Of course, the important thing for any such estimation model is how applicable it is in everyday work. And of course the model can be further extended with new varieties.

Given that the statistical properties of the estimation models are good, they are all set up for the right sample size - so it is fair to say that they will be suitable for this year's samples.

Our work could be the basis for further categories, but this will of course require more samples and more measurements to be added to the current database

 Yours sincerely,

Marietta Fodor

Reviewer 4 Report

The manuscript is prepared quite well and interesting for a broad readership. Even though, manuscript on a very similar topic and the same type of sample have been published before, this work can be seen as a further advancement extending the existing knowledge about plum quality control.

The manuscript is recommended for considering the comments provided below.

- Band assignment should be provided in more details

- Regression vector analysis should be added

Author Response

Assessment of maturity of plum samples using FT-NIR technique combined with chemometric methods

by

Marietta Fodor, Zsuzsa Jókai, Anna Matkovits, Eszter Benes

Response to Reviewer 4

Thank you for your time to review and thank you for your comment.

The manuscript is prepared quite well and interesting for a broad readership. Even though, manuscript on a very similar topic and the same type of sample have been published before, this work can be seen as a further advancement extending the existing knowledge about plum quality control.

The manuscript is recommended for considering the comments provided below.

- Band assignment should be provided in more details

We did not go into the vibration areas in more detail because there are countless literature materials available on this issue. That's why we have selected and named only the typical areas.

- Regression vector analysis should be added

For regression vector analysis, we originally provided the accuracy, precision, sensitivity and specificity results in the manuscript. In the supplement we report the results of the preliminary ANOVA test.

We would like to request clarification on what specifically the Reviewer meant.

Yours sincerely,

Marietta Fodor
